# Suppression of STAT3 Phosphorylation and RelA/p65 Acetylation Mediated by MicroRNA134 Plays a Pivotal Role in the Apoptotic Effect of Lambertianic Acid

**DOI:** 10.3390/ijms20122993

**Published:** 2019-06-19

**Authors:** Deok Yong Sim, Hyo-Jung Lee, Ji Hoon Jung, Eunji Im, Jisung Hwang, Dong Sub Kim, Sung-Hoon Kim

**Affiliations:** College of Korean Medicine, Kyung Hee University, Seoul 02447, Korea; Simdy0821@naver.com (D.Y.S.); hyonice77@naver.com (H.-J.L.); johnsperfume@khu.ac.kr (J.H.J.); ji4137@naver.com (E.I.); hjsung0103@naver.com (J.H.); dongsoub@hanmail.net (D.S.K.)

**Keywords:** Lambertianic acid, STAT3, NF-κB, p300, Ac-RelA/p65

## Abstract

As p300-mediated RelA/p65 hyperacetylation by signal transducers and activators of transcription 3 (STAT3) is critical for NF-κB activation, in the current study, the apoptotic mechanism of lambertianic acid (LA) was explored in relation to STAT3 phosphorylation and RelA/p65 acetylation in MCF-7, DU145, PC-3, and MDA-MB-453 cells. LA significantly increased the cytotoxicity, sub G 1 population, and the cleavage of poly (ADP-ribose) polymerase (PARP) in MDA-MB-453 or PC-3 cells (STAT3 mutant), more than in the MCF-7 or DU145 cells (STAT3 wild). Consistently, LA inhibited the phosphorylation of STAT3 and nuclear factor kappa-light-chain-enhancer of activated B cells (NF-κB), and disrupted the interaction between p-STAT3, p300, NF-κB, and RelA/p65 acetylation (Ac-RelA/p65) in the MCF-7 and DU145 cells. Also, LA reduced the nuclear translocation of STAT3 and NF-κB via their colocalization, and also suppressed the protein expression of XIAP, survivin, Bcl-2, Bcl-xL, vascular endothelial growth factor (VEGF), Cox-2, c-Myc and mRNA expression of interleukin 6 (IL-6), and tumor necrosis factor-α (TNF-α) in MCF-7 cells. Conversely, IL-6 blocked the ability of LA to suppress the cytotoxicity and PARP cleavage, while the depletion of STAT3 or p300 enhanced the PARP cleavage of LA in the MCF-7 cells. Notably, LA upregulated the level of miRNA134 and so miRNA134 mimic attenuated the expression of pro-PARP, p-STAT3, and Ac-RelA, while the miRNA134 inhibitor reversed the ability of LA to reduce the expression of Ac-RelA and pro-PARP in MCF-7 cells. Overall, these findings suggest that LA induced apoptosis via the miRNA-134 mediated inhibition of STAT3 and RelA/p65 acetylation.

## 1. Introduction

Signal transducers and activators of transcription 3 (STAT3), a member of a family of seven proteins (STATs 1, 2, 3, 4, 5a, 5b, and 6), is involved in proliferation, angiogenesis, metastasis, and immunosuppression [1]. Interleukin 6 (IL-6) activates STAT3 via the phosphorylation of tyrosine 705 [2], while STAT3 inhibitors, such as STA-21 [3], Stattic [4,5], S31-201 [6], and BP-1-102 [7], inhibit the phosphorylation of STAT3 [8] via interaction with the STAT3 SH-2 domains.

Accumulating evidence has revealed that p300-mediated RelA/p65 hyperacetylation by STAT3 is essential for NF-κB activation in several cancers [9], as protein hyper-acetylation in cells is activated in inflammation and cancers [10]. In the same line, acetylation, like phosphorylation, is important in regulating the nuclear translocation of NF-κB [11].

Hence, recently several research projects have been conducted using natural compounds targeting the hypoacetylation of p65 by directly inhibiting the activity of p300 histone acetyltransferase (HAT) enzymes (acetyltranferase) [12,13]. Indeed, anacardic acid [14], gallic acid [13], and epigallocatechin gallate (EGCG) [12] suppressed NF-κB and its regulated gene products by inhibiting p65 acetylation.

MicroRNAs (miRNAs), 20–22 nucleotide non-coding RNAs, were known to work mainly as post-transcriptional regulators of mRNAs [15]. Among them, miR134 suppresses the proliferation and epithelial–mesenchymal transition (EMT) in colorectal and renal cancers [16], as a tumor suppressor through the KRAS-mediated mitogen-activated protein kinase (MAPK)/ extracellular-signal-regulated kinase (ERK) pathway [17].

Lambertianic acid (LA; Figure 1a), a major component of *Pinus koraiensis*, was known to have anti-obesity [18], anti-inflammatory [19], and anti-cancer effects [20,21,22]. Nevertheless, the underlying molecular mechanisms of LA were not fully understood in association with STAT3 phosphorylation and RelA/p65 acetylation. Hence, in the present work, the apoptotic mechanism of LA was examined in the MCF-7 and DU145 (STAT3 wild type), MDA-MB-453 (STAT3 mutant), and PC-3(STAT3 null) cell lines [23], in association with the miR134 mediated suppression of STAT3 and RelA/p65 acetylation. 

## 2. Results

### 2.1. LA Induced Cytotoxicity and Sub-G1 Accumulation Increased the Cleavage of Poly (ADP-Ribose) Polymerase (PARP) in STAT3-Dependent or STAT3-Independent Cancer Cells

To evaluate the specific apoptotic effect of LA in STAT3-dependent or independent cancer cells, a cytotoxicity assay was conducted in breast cancer (MDA-MB-453; STAT3 mutant, MCF-7; STAT3 wild type) and prostate cancer (PC-3; STAT3 null, DU145; STAT3 wild type) cell lines by 3-(4,5-dimethylthiazol-2-yl)-2,5diphenyltetrazolium bromide (MTT) assay. Here, LA reduced the viability of the DU145, PC-3 cells, MCF-7, and MDA-MB-453 cells in a dose-dependent manner (Figure 1b). However, the cytotoxicity of LA was better in the MDA-MB-453 and PC-3 cells than in the MCF-7 and DU145 cells. Consistently, the cell cycle analysis revealed that LA at 30 μM increased the sub-G1 population to 42.75% in MDA-MB-453, more than 6.84% of the MCF-7 cells (Figure 1c). Also, LA enhanced the cleavages of PARP in the MDA-MB-453 and PC-3 cells better than in the MCF-7 and DU145 cells (Figure 1d).

### 2.2. LA Suppressed the Phosphorylation of STAT3 and NF-κB, and the Expression of p300 and RelA Acetylation in MCF-7 and DU145 Cells

To determine the role of STAT3 in LA-induced apoptosis, Western blotting was conducted in MCF-7 and DU145 cells. As shown in Figure 2a, LA attenuated the phosphorylation of STAT3 and NF-κB, and also reduced the expression of p300 and RelA/p65 acetylation in MCF-7 and DU145 cells. Consistently, LA effectively suppressed the nuclear translocation of p-STAT3 and NF-κB p65 via their co-localization in MCF-7 cells (Figure 2b).

### 2.3. LA Attenuated the Expression of NF-κB Regulated Genes in MCF-7 Cells

To confirm the p65 acetylation inhibition of LA, Western blotting and qRT-PCR were conducted in MCF-7 cells. Consistently, the LA attenuated the expression of the NF-κB regulated genes, including Bcl-2, Bcl-xL, XIAP, survivin, (anti-apoptotic proteins), VEGF (angiogenic protein), COX-2 (inflammatory protein), and c-Myc (oncogenic genes) in the MCF-7 cells (Figure 3a). Also, STAT3 activators IL-6 and TNF-α (inflammatory factor) were downregulated at the mRNA level in the LA-treated MCF-7 cells (Figure 3b).

### 2.4. STAT3 Inducer IL-6 Suppressed Cytotoxic and Apoptotic Effects of LA in MDA-MB-453 Cells

To verify the critical role of STAT3 in the cytotoxic and apoptotic effect of LA, MTT assay, Western blotting, and cell cycle analysis were conducted in IL-6-treated MDA-MB-453 cells. Here, IL-6 disturbed the cytotoxicity of LA (Figure 4a), and also reduced the sub-G1 population to 12.11% compared to the LA-alone treated control (22.43%) in IL-6 treated MDA-MB-453 cells (Figure 4c). Consistently, the IL-6 treatment blocked the capacity of LA to induce PARP cleavage (Figure 4b); decrease the phosphorylation of STAT3, IkB, and p65; and reduce the expression of p300 and Ac-RelA in MDA-MB-453 cells compared to the untreated control (Figure 4d). Conversely, the depletion of STAT3 or p300 enhanced the PARP cleavage and the inhibition of Ac-RelA A by LA in the MCF-7 cells (Figure 4e,f).

### 2.5. LA Inhibited the Expression of p300 and RelA/65 Acetylation, and Disrupted the Interaction between p-STAT3, p300, and NF-κB in MCF-7 and DU145 Cells

To validate whether LA inhibits the p300-mediated RelA/p65 acetylation, immunofluorescence and co-immunoprecipitation (IP) were performed in MCF-7 and DU145 cells. As shown Figure 5a, the scores of the protein–protein interactions (PPI) between STAT3 and p300, STAT3 and p65, or p65 and p300 were 0.970, 0.820, and 0.981, respectively. IP revealed that LA disrupted the binding between p-STAT3, p300, and RelA (p65) after treatment with LA for 24 h, in the MCF-7 and DU145 cells (Figure 5b). 

### 2.6. miR134 Plays a Pivotal Role in the LA-Induced Apoptotic Effect in MCF-7 Cells

To identify the role of miR134 in the LA-induced apoptotic effect in MCF-7 cells, the miR134-mimic or -inhibitor plasmid was transfected into MCF-7 cells. Here, LA increased the expression of miR134 in the MCF-7 cells by RT-PCR (Figure 6a). The miR134-mimic attenuated the expression of the pro-PARP, p-STAT3, and Ac-RelA in MCF-7 cells (Figure 6b). In contrast, the miR134 inhibitor reversed the cytotoxicity and inhibition of pro-PARP and Ac-RelA by LA in the MCF-7 cells (Figure 6c,d).

## 3. Discussion

The aim of the current work is to elucidate the underlying apoptotic mechanism of LA in association with STAT3 and NF-κB signaling in breast and prostate cancer cells. Herein, MDA-MB-453 and PC-3 STAT3 mutant cells were more susceptible to the cytotoxicity of LA, compared with MCF-7 and DU145 STAT3 wild-type cells. Likewise, LA significantly increased sub-G1 accumulation along with an increased cleavage of PARP in the MDA-MB-453 and PC-3 cells, more than in the MCF-7 and DU145 cells, implying the important role of STAT3 in the cytotoxic and apoptotic effect of LA.

Accumulating evidence has revealed that STAT3 and NF-κB are involved in a variety of biological processes, such as inflammation, and cancer progression and growth [24,25]. Thus, STAT3 and NF-κB can be considered as potent target molecules for cancer therapy [26,27]. Our Western blotting showed that LA treatment suppressed the phosphorylation of STAT3 and NF-κB, and the nuclear translocation of STAT3 and NF-κB, indicating the involvement of STAT3 and NF-κB pathways in LA-induced apoptosis in MCF-7 and DU145 cells.

Recent evidence has demonstrated that STAT3 is upregulated in most cancers, and NF-κB enhances the STAT3-p300 interaction [9]. Also, the accumulation of RelA/p65 in the nucleus can be promoted by acetylation by p300 [28], and STAT3 is essential for p300-mediated RelA acetylation [29,30]. Through the close interaction between STAT3 and NF-κB, STAT3 increases the NF-κB activity, and also, the persistent activation of STAT3 is dependent on NF-κB signaling [9,31]. It is well documented that several inflammatory cytokines, including IL-6, COX-2, and IL-23, activate STAT3 through NF-κB regulated inflammatory responses, and RelA/p65 is maintained through p300-mediated RelA/p65 acetylation by STAT3 [29]. Here, LA attenuated the expression of p300 and RelA/p65 acetylation (Ac-RelA), and also disrupted the binding of p-STAT3 with p300 or Ac-RelA in MCF-7 and DU145 cells, indicating that LA induces the hypoacetylation of RelA/p65, leading to the downregulation of NF-κB-regulated proteins via the interrupted binding of p-STAT3 with p300 or NF-κB. 

As p300 HAT is closely associated with RelA/p65 acetylation in cancer progression [32,33], several dietary compounds, such as anacardic acid [34], garcinol [35], curcumin [36], gallic acid [13], and EGCG [12], are reported to inhibit p300. Importantly, anacardic acid suppressed the expression of the NF-κB-regulated gene products by inhibiting p65 acetylating [37]. Similarly, silencing p300 using its siRNA plasmid transfection significantly enhanced RelA deacetylation and apoptosis induced by LA, implying a critical role of p300 in the apoptotic effect of LA. 

Furthermore, LA suppressed NF-κB-regulated genes, including anti-apoptotic proteins (Bcl-2, Bcl-xL, XIAP, and Survivin), angiogenic and proliferative proteins (VEGF, Cox-2, and c-Myc), and inflammatory factors (IL-6 and TNF-α), in MCF-7 cells. 

MicroRNAs are well known to mediate cell differentiation, proliferation, and apoptosis [38]. Recent evidence revealed that miR134 induces apoptosis, and inhibits proliferation and migration by targeting the STAT3 in bladder cancer cells [39], and miR134 abrogates proliferation and EMT in renal cell carcinoma and colorectal cancer cells [17]. Here, LA upregulated the miRNA level of miRNA134 in MCF-7 cells, and also, the miRNA134-mimic abrogated the expression of p-STAT3, pro-PARP, and Ac-RelA, whereas the miRNA134-inhibitor reversed the ability of LA to inhibit the expression of pro-PARP and Ac-RelA in MCF-7 cells, indicating a critical role of miRNA134 in LA-induced apoptosis.

In summary, LA significantly increased the cytotoxicity, sub-G 1 population, and cleavage of PARP in MDA-MB-453, PC-3, MCF-7, and DU145 cells. Also, LA reduced the phosphorylation of STAT3 and NF-κB, and disrupted the interaction between p-STAT3, p300, NF-κB, and RelA/p65 acetylation in the MCF-7 and DU145 cells. Furthermore, LA reduced the nuclear translocation of STAT3 and NF-κB, and suppressed several survival genes, including Bcl-2, Bcl-XL, XIAP, survivin, VEGF, Cox-2, C-Myc, and IL-6, and TNF-α in MCF-7 cells. However, IL-6 blocked the apoptotic effect of LA, and the depletion of the STAT3- or p300-enhanced PARP cleavage of LA in MCF-7 cells. Notably, LA activated miRNA134, and so the miRNA134-mimic reduced the expression of p-STAT3, pro-PARP, and Ac-RelA, while the miRNA134 inhibitor reversed the apoptotic effect of LA in the MCF-7 cells. Taken together, these findings demonstrate that LA induces apoptosis via the miRNA-134 mediated inhibition of STAT3 and RelA/p65 acetylation (Figure 7).

## 4. Materials and Methods

### 4.1. Isolation of Lambertianic Acid (LA)

As shown in our papers [16,18], lambertianic acid (LA; molecular weight = 316.43) was isolated, purified, and identified as having over 98% purity, based on spectroscopic analyses such as nuclear magnetic resonance (NMR), mass spectrometry (MS), and infrared (IR) [40].

### 4.2. Cell Culture

Human breast cancers (MCF-7 (ATCC^®^ HTB-22^™^) and MDA-MB-453 (ATCC^®^ HTB-131^™^)) and prostate cancer (DU145 (ATCC^®^ HTB-81^™^) and PC-3 (ATCC^®^ CRL-1435^™^)) cells were obtained from the American Type Culture Collection (ATCC; Manassas, VA, USA). These cells were cultured in RPMI1640 supplemented with 10% fetal bovine serum (FBS) and 1% antibiotic (Welgene, Inc., Gyeongsan, South Korea) at 37 ˚C, under a humid environment with 5% CO_2_.

### 4.3. Cell Viability Assay

The cytotoxicity of LA was evaluated by using a 3-(4,5-dimethylthiazol-2-yl)-2,5diphenyltetrazolium bromide (MTT) assay. Briefly, the MCF-7, MDA-MB-453, DU145, and PC-3 cells (1 × 10^4^ cells/well) were exposed to indicated concentrations of LA for 24 h, and were incubated with MTT (1 mg/mL); (Sigma Chemical, St. Louis, MO, USA) for 2 h. Then, the cell viability was calculated as a percentage of the viable cells in the LA-treated group vs. the untreated control, with optical density (OD) values obtained using a microplate reader (Molecular Devices, LLC, Sunnyvale, CA, USA) at 570 nm.

### 4.4. Cell Cycle Analysis

Based on Lee et al.’s paper [20], MCF-7 and MDA-MB-453 cells (2 × 10^5^ cells/mL) were exposed to LA (0, 15, and 30 μM) for 24 h, incubated with RNase A (10 mg/mL) for 1 h at 37 °C, and stained with propidium iodide (50 μg/mL). The stained cells were calculated for the DNA content by FACSCalibur (Becton Dickinson, Franklin Lakes, NJ, USA) using CellQuest Software.

### 4.5. Western Blotting

The cells were exposed to various concentrations of LA for 24 h; lyzed in a lysis solution (50 mM Tris–HCl, pH 7.4, 150 mM NaCl, 1% Triton X-100, 0.1% Sodium Dodecyl Sulfate (SDS), 1 mM EDTA, 1 mM Na_3_VO_4_, 1 mM NaF, and 1× protease inhibitor cocktail) on ice; and spun down at 14,000× *g* for 20 min at 4 °C. The supernatants were collected and quantified for protein concentration using an reducing agent compatible (RC) detergent compatible (DC) protein assay kit (Bio-Rad, Hercules, CA, USA). The protein samples were separated on 4%–12% NuPAGE Bis–Tris gels (Novex, Carlsbad, CA, USA), and transferred to a Hybond ECL transfer membrane for detection with antibodies for cleaved-PARP, p-JAK2, p-STAT3, STAT3, p- p65/RelA, p65/RelA, p-IκB, p300, Ac-RelA, Bcl-2, Bcl-xL, XIAP, survivin, VEGF, COX-2, and c-Myc (Cell Signaling Technology, Beverly, MA, USA), as well as Bcl-2, XIAP, and COX-2 (Santa Cruz Biotechnology, Inc., Dallas, TX, USA), and β-actin (Sigma, St. Louis, MO, USA). The protein expression was measured using an enhanced chemiluminescence (ECL) system (Amersham Pharmacia, Piscataway, NJ).

### 4.6. Immunofluorescence

The cells fixed on a poly-L-lysine-coated slide in 4% paraformaldehyde were permeabilized in 0.1% Triton X-100, followed by immunostaining with goat polyclonal anti-phospho-STAT3 and rabbit polyclonal anti-p65 (Santa Cruz Biotechnology, Inc., Dallas, TX, USA) antibodies, using Goat and Rabbit IgG fluorescein isothiocyanate (FITC) antibody H&L as a secondary antibody. The cells were mounted in a medium containing 4′,6-diamidino-2-phenylindole (DAPI) and were photographed under an FLUOVIEW FV10i confocal microscopy (Olympus Corporation, Tokyo, Japan).

### 4.7. Co-Immunoprecipitation

The MCF-7 and DU145 cells were lyzed and immunoprecipitated with STAT3 and p300 antibodies. Thereafter, protein A/G sepharose beads (Santa Cruz Biotechnology, Santa Cruz, CA) were applied. The final precipitated proteins were subjected to immunoblotting with the indicated antibodies.

### 4.8. RNA Interference

The plasmids for p300-siRNA or control-siRNA (40 nM; Bioneer, Daejeon, Korea) were transiently transfected into MCF-7 cells for 24 h, using an INTERFERinTM transfection reagent (Polyplus- transfection Inc., New York, NY), according to manufacturer’s protocols.

### 4.9. RT-qPCR Analysis

As shown in Matsuno et al.’s paper, the total RNA from the LA-treated MCF-7 cells was isolated by QIAzol (Invitrogen, Carlsbad, CA, USA) and subjected to regular processes of RT-qPCR using the following primers, IL-6-forward: 5′-CCACCGGGAACGAAAGAGAA-3′; reverse-5′-GAGAAGGCAACTGGACCGAA -3′ (Bioneer, Daejeon, Korea), TNF- α- forward: 5′- GCCGCATCGCCGTCTCCTAC-3′; reverse- 5′-CCTCAGCCCCTCTGGGGTC -3′ (Bioneer, Daejeon, Korea), hGAPDH-forward5′-CCA CTC CTC CAC CTT TGA CA-3′;reverse-5′-ACC CTG TTG CTG TAG CCA -3′ (Bioneer, Daejeon, Korea). The mRNA level of glyceraldehyde-3-phosphate dehydrogenase (GAPDH) was used to normalize the expression of the genes of interest. Each sample was tested in triplicate, and the relative gene expression data were analyzed by means of the 2^−ΔCT^ method.

### 4.10. MicroRNA Transfection Assay

miR134-mimic, miR134-inhibitor, and miR-Con(200nM) plasmids (Bioneer, Daejeon, Korea) were transfected intoMCF-7 cells using an X-tremeGENE HP DNA Transfection Reagent (Roche, Basel, Switzerland), according to the manufacture’s protocols.

### 4.11. Statistical Analysis

All of the values were expressed as means ± standard deviation (SD). Sigmaplot version 12 software (Systat Software Inc., San jose, CA, USA) was used for the statistical analysis. Student *t*-test was used for comparison of two groups, and *p* < 0.05 was considered as statistically significant. 

## 5. Conclusions

Our findings suggest that LA significantly increased the cytotoxicity, sub G 1 population, and the cleavage of PARP in MDA-MB-453, PC-3, MCF-7, and DU145 cells. Also, LA reduced the phosphorylation of STAT3 and NF-κB, and disrupted the interaction between p-STAT3, p300, NF-κB, and RelA/p65 acetylation in MCF-7 and DU145 cells. Furthermore, LA reduced the nuclear translocation of STAT3 and NF-κB, and suppressed several survival genes, including Bcl-2, Bcl-XL, XIAP, survivin, VEGF, Cox-2, C-Myc, IL-6, and TNF-α in MCF-7 cells. However, IL-6 blocked the apoptotic effect of LA and the depletion of the STAT3- or p300-enhanced PARP cleavage of LA in MCF-7 cells. Notably, LA activated miRNA134, and so the miRNA134 mimic attenuated the expression of p-STAT3, pro-PARP, and Ac-RelA, while the miRNA134 inhibitor reversed the apoptotic effect LA in the MCF-7 cells. Taken together, these findings demonstrate that LA induces apoptosis via the miRNA-134 mediated inhibition of STAT3 and RelA/p65 acetylation

## Figures and Tables

**Figure 1 ijms-20-02993-f001:**
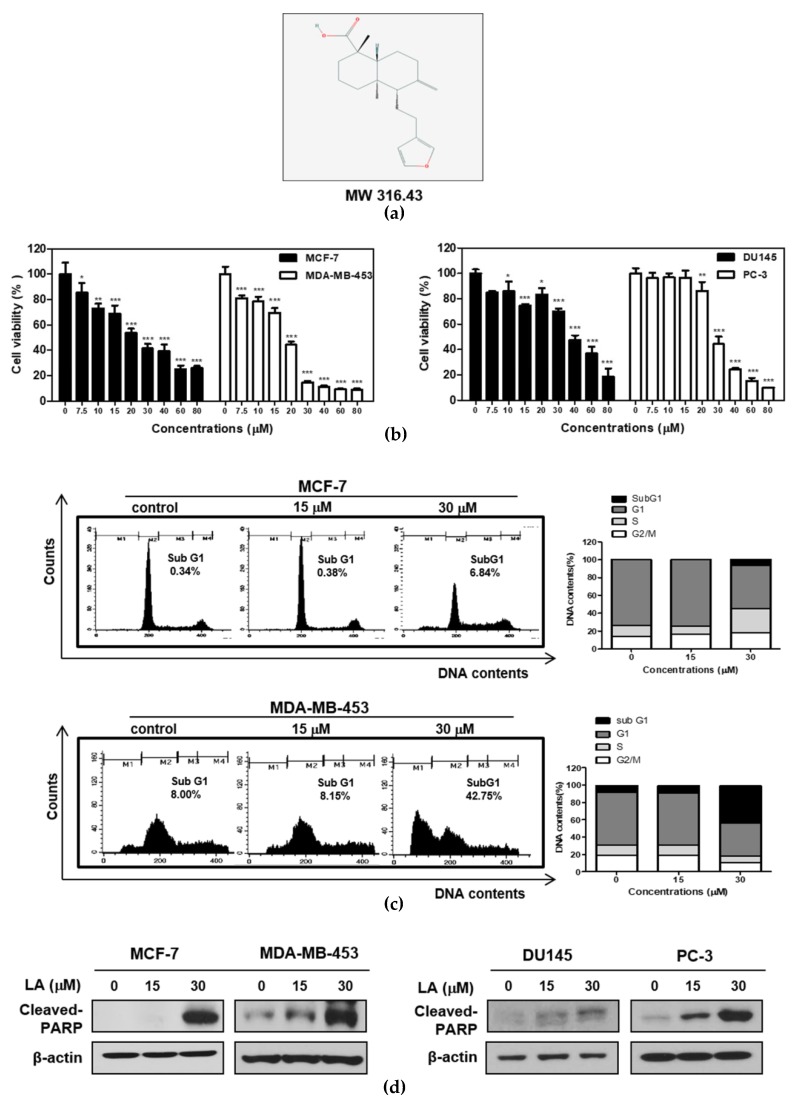
Effect of lambertianic acid (LA) on cytotoxicity and apoptosis in MCF-7, MDA-MB-453, DU145, and PC-3 cancer cells. (**a**) The chemical structure of LA. (**b**) The effect of LA on cytotoxicity in MCF-7, MDA-MB-453, DU145, and PC-3 cells. The cells were distributed onto 96-well plates and were exposed to indicated concentrations of LA (0, 7.5, 10, 15, 20, 30, 40, 60, and 80 μM) for 24 h. The cell viability was calculated using a 3-(4,5-dimethylthiazol-2-yl)-2,5diphenyltetrazolium bromide (MTT) assay. The data stand for means ± standard deviation (SD). * *p* < 0.05, ** *p* < 0.01, *** *p* < 0.001. (**c**) Effect of LA on the sub-G1 population in MCF-7 and MDA-MB-453 cells, by cell cycle analysis. The MCF7 and MDA-MB-453 cells were exposed to LA (0, 15, and 30 μM) for 24 h, and were stained with propidium iodide (PI) for flow cytometric analysis. The bar graphs represent the quantification of the cell cycle population (%). (**d**) The effect of LA on the poly (ADP-ribose) polymerase (PARP) cleavage in the MCF-7 and MDA-MB-453 cells. The MCF-7 and MDA-MB-453 cells were treated with LA (0, 15, and 30 μM) for 24 h, and were subjected to Western blotting with the antibody of the cleaved PARP.

**Figure 2 ijms-20-02993-f002:**
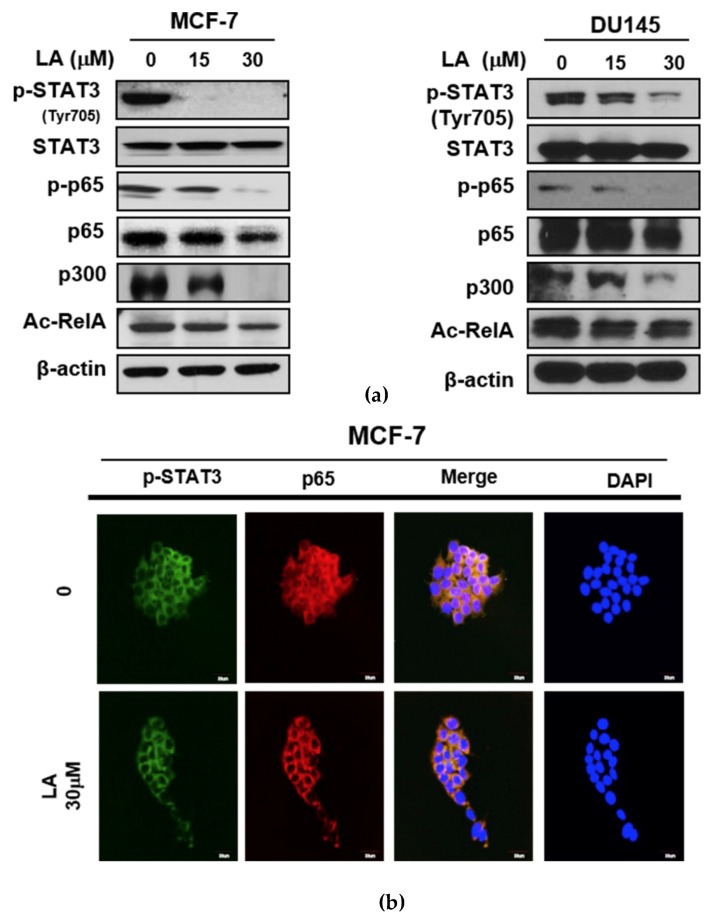
Effect of LA on the expression of the p-signal transducers and activators of transcription 3 (STAT3), p-NF-κB, p300, and Ac-RelA, and their nuclear translocation in MCF-7 and DU145 cells. (**a**) Effect of LA on p-STAT3, p-NF-κB, p300, and Ac-RelA in MCF-7 and DU145 cells. The cells were treated with LA (0, 15, and 30 μM) for 24 h, and were subjected to Western blotting for p-JAK2, p-STAT3, STAT3, p-p65/RelA, NF-κB. p300, and Ac-RelA. (**b**) Effect of LA on nuclear translocation in the MCF-7 cells. Immunostaining was conducted with antibodies of p-STAT3 and RelA/p65, and secondary fluorescein isothiocyanate (FITC)-conjugated antibody in the MCF-7 cells treated with or without LA.

**Figure 3 ijms-20-02993-f003:**
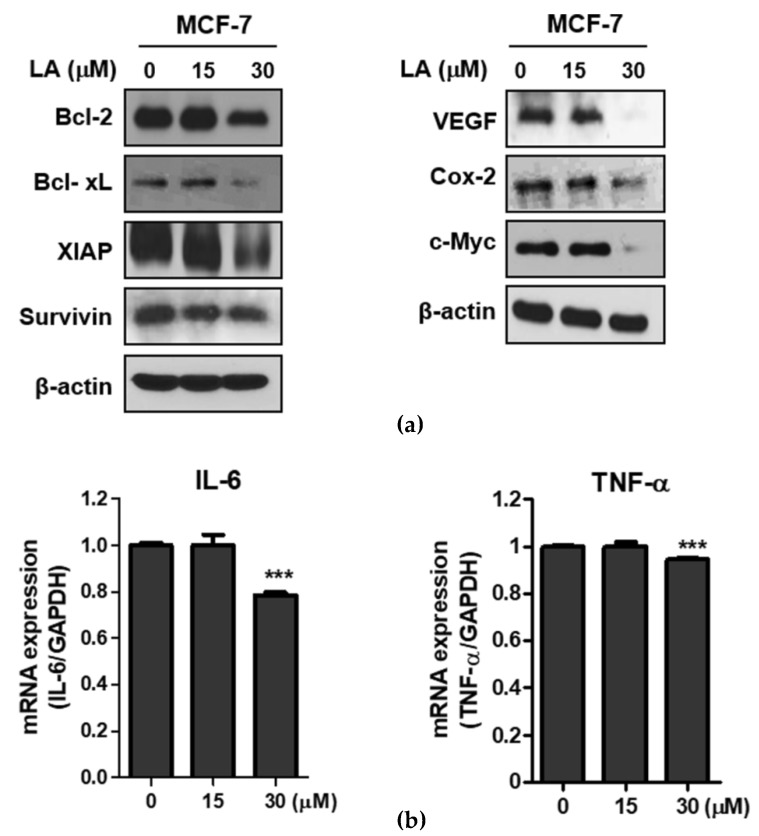
Effect of LA on NF-κB-regulated proteins in MCF-7 cells. (**a**) Effect of LA on NF-κB-modulated genes in MCF-7 cells. The MCF-7 cells were treated with LA (0, 15, and 30 μM) for 24 h, and were subjected to Western blotting for Bcl-2, Bcl-xL, XIAP, survivin, VEGF, COX-2, c-Myc, and β actin. (**b**) Effect of LA on TNF-α and interleukin 6 (IL-6) at the mRNA level in the MCF-7 cells by qRT-PCR. Data stand for means ± SD. *** *p* < 0.001 vs. untreated control.

**Figure 4 ijms-20-02993-f004:**
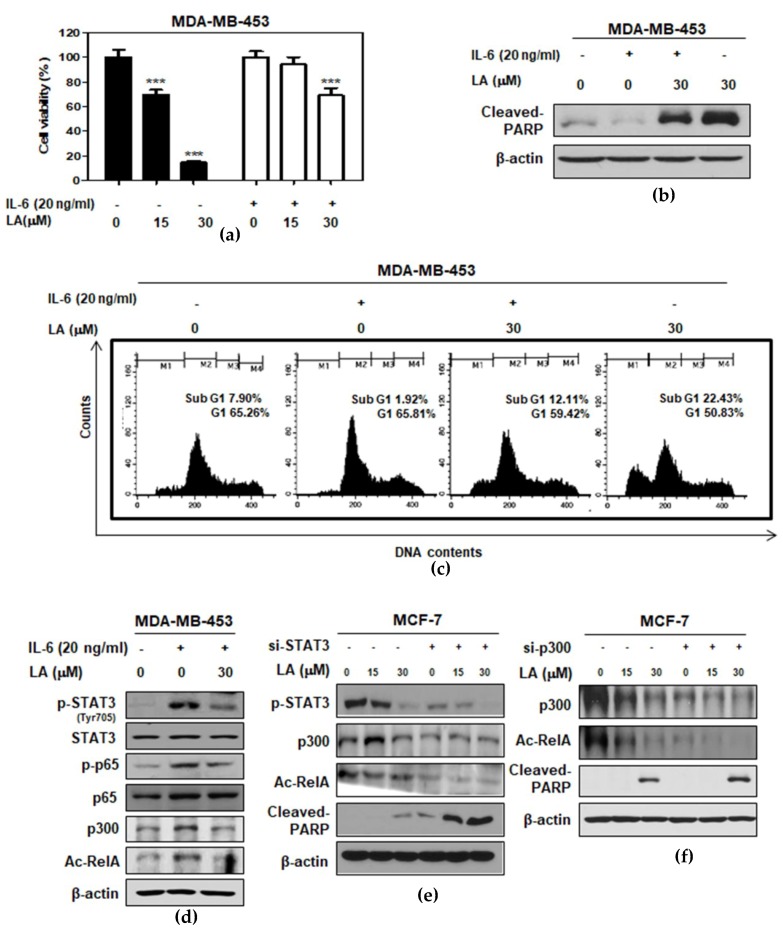
Effect of IL-6 or the depletion of STAT3 or p300 on the cytotoxic and apoptotic effects by LA in MCF-7 and MDA-MB-453 cells. (**a**) Effect of LA on the cytotoxicity of LA in IL-6-treated MDA-MB-453 cells. The cells were exposed to LA (0, 15, and 30 μM) for 24 h, with or without IL-6 (20 ng/mL). The cell viability was calculated by 3-(4,5-dimethylthiazol-2-yl)-2,5diphenyltetrazolium bromide (MTT) assay. (**b**) Effect of LA on the PARP cleavage in the IL-6-treated MDA-MB-453 cells. The cells were exposed to LA (30 μM) for 6 h, with or without IL-6 (20 ng/mL). Cell lysates were prepared and subjected to Western blotting for PARP. (**c**) The effect of LA on the sub-G1 population in the IL-6-treated MDA-MB-453 cells. The cells were exposed to LA (30 μM) for 6 h, with or without IL-6 (20 ng/mL) stimulation, and cell cycle analysis was conducted. (**d**) The cell lysates prepared from the MDA-MB-453 cells were subjected to Western blotting for p-STAT3, STAT3, and p-p65/RelA. (**e**) The effect of STAT3 depletion on p-STAT3, p-p65, p300, Ac-RelA, and cleaved PARP in the LA-treated MCF-7 cells. The cells transfected with p-STAT3 or scrambled siRNA plasmid (40 nM) for 24 h were exposed to LA (30 μM) for 24 h, and were subjected to Western blotting for p-STAT3, STAT3, p65, p300, and Ac-RelA. (**f**) The effect of p300 depletion on p300, Ac-RelA, and cleaved PARP in the LA-treated MCF-7 cells. The cells transfected with either p300 or scrambled siRNA plasmid (40 nM) for 24 h were exposed to LA (0, 15, and 30 μM) for 24 h, and subjected to Western blotting with antibodies of p300, Ac-RelA, and cleaved-PARP.

**Figure 5 ijms-20-02993-f005:**
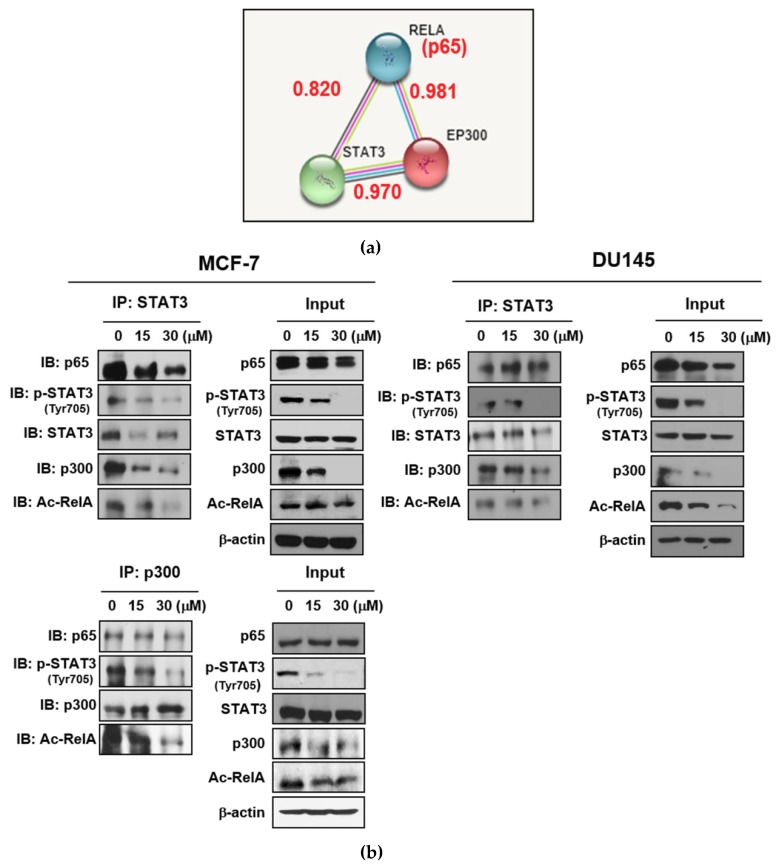
Effect of LA on the interaction between p-STAT3, p300, and RelA in the MCF-7 and DU145 cells. (**a**) Protein–protein interaction (PPI) scores between STAT3, p300, and RelA/p65 by search tool for the retrieval of interacting genes/proteins (STRING) database. (**b**) MCF-7 and DU145 cells were treated with LA for 24 h. Immunoprecipitation (IP) was performed with cell lysates, using antibodies of STAT3 and p300, and Western-blot analysis was conducted to detect RelA/p65, p-STAT3, STAT3, p300, and Ac-RelA in the whole cell lysates. IB—immunoblotting.

**Figure 6 ijms-20-02993-f006:**
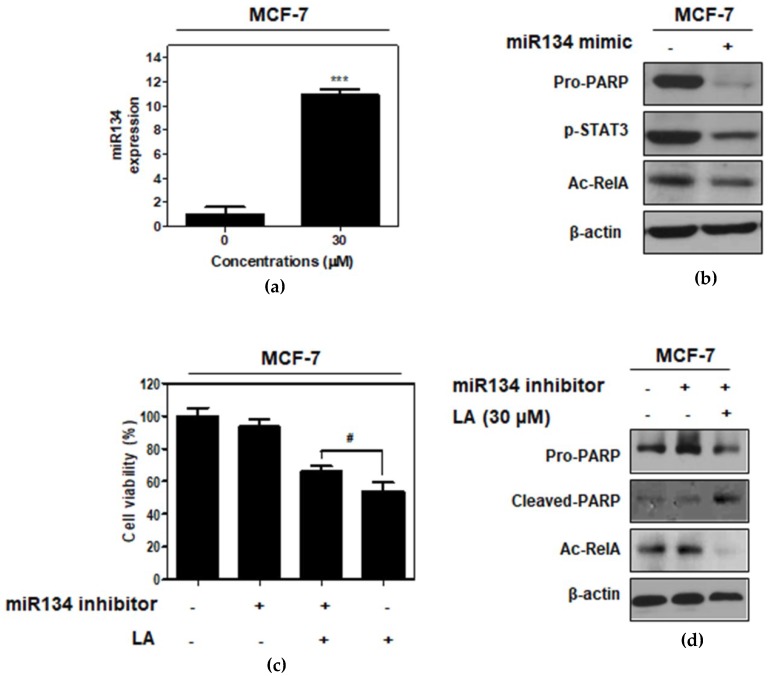
The pivotal role of miR134 in the LA-induced apoptotic effect in MCF-7 cells. (**a**) Effect of LA on the miR134 level in MCF-7 cells by qRT-PCR. *** *p* < 0.001 vs. untreated control. (**b**) Effect of miR134 mimic on PARP, p-STAT3, and Ac-RelA in the MCF-7 cells. (**c**) Effect of the miR134-inhibitor on the viability of the MCF-7 cells with or without LA. # *p* < 0.05 vs. LA alone treated control. (**d**) Effect of the miR134-inhibitor on PARP and Ac-RelA in the MCF-7 cells. Data represent means ± SD.

**Figure 7 ijms-20-02993-f007:**
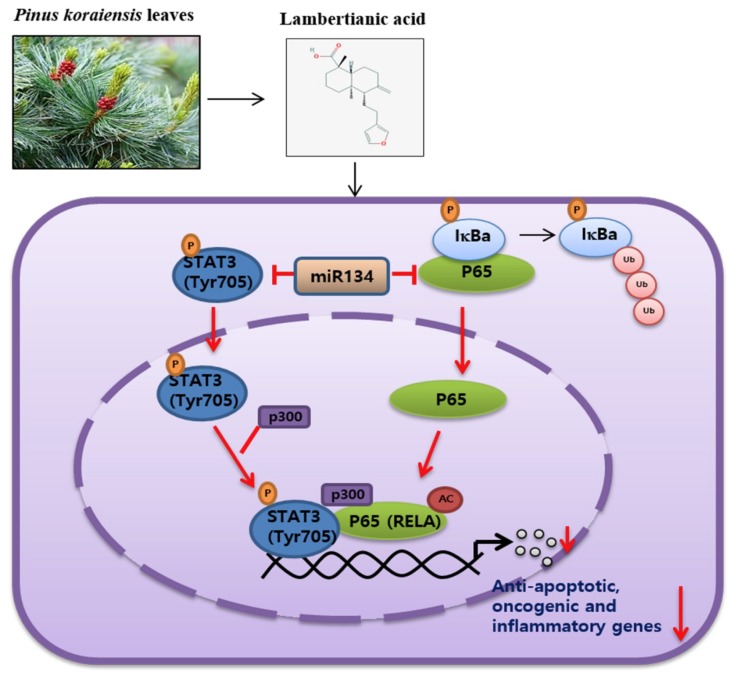
Schematic diagram on the apoptotic mechanism of LA in cancer cells via the miRNA-134-mediated inhibition of STAT3 and RelA/p65 acetylation.

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
