# Peer review of "Suppression of STAT3 Phosphorylation and RelA/p65 Acetylation Mediated by MicroRNA134 Plays a Pivotal Role in the Apoptotic Effect of Lambertianic Acid"

_ijms, 2019, doi:10.3390/ijms20122993_

Reviewer 1 Report

IJMS-525399

The authors examined effects of lambertianic acido on STAT3 phosphorylation and RelA3 /p65 acetylation mediated by microRNA134.

The method and results can be improved. However, the findings may be interesting for the readers. Therefore, I could recommend to accept the manuscript for publication in International Journal of Molecular Sciences as Article, after minor revisions on the following basis.

1. At first, this manuscript should get some English proofreading, including figure captions.

2. in Introduction, STAT6 should be added. (STAT1,2,3,4,5a,5b, )

3. In Abstract, LA should not to be described as abbreviation, in first time.

4. In 2.4. section, please unify significant figures of G1%.

5. In Fig.5, what “IB” means should be explained.

Author Response

Reviewer: 1
Open Review

Comments and Suggestions for Authors

The authors examined effects of lambertianic acido on STAT3 phosphorylation and RelA3 /p65 acetylation mediated by microRNA134.

The method and results can be improved. However, the findings may be interesting for the readers. Therefore, I could recommend to accept the manuscript for publication in International Journal of Molecular Sciences as Article, after minor revisions on the following basis.

 1. At first, this manuscript should get some English proofreading, including figure captions.

 (Response) Thanks. Carefully edited.

2. in Introduction, STAT6 should be added. (STAT1,2,3,4,5a,5b, )

 (Response) Thanks for your valuable comment. Added.

3. In Abstract, LA should not to be described as abbreviation, in first time.

 (Response)Thanks. Corrected.

4. In 2.4. section, please unify significant figures of G1%.

  (Response) Thanks. G1% was added in Figure 4C.

5. In Fig.5, what “IB” means should be explained.

  (Response) Thanks. The IB means “Immunoblotting”

Reviewer 2 Report

This is an interesting study showing the effects of lambertianic acid (LA) on apoptotic mechanism in a variety of cancer cells (MCF-7, DU145 and MDA-MB-453 cell lines). The authors concluded that LA induced apoptosis via miRNA-134 mediated inhibition of STAT3 and RelA/p65 acetylation. Technically the study is well done and the conclusions are supported by the data. However, a number of issues should be addressed by the authors.

qRT-PCR analyses should be detailed. The authors only have included the primers. How they analyzed data and comparisons… Also for western blots.

LA attenuated the expression of NF-B regulated genes in MCF-7 cells. Did the authors investigate this point in other cells?

The effect of miR134 inhibitor on the viability of MCF-7 cells with or without LA in fig 6 seems to show that miR134 did not produce any effect on LA treated cells despite affecting pro-PARP and Ac-RelA. This point should be discussed.

Author Response

Reviewer: 2

Comments and Suggestions for Authors

This is an interesting study showing the effects of lambertianic acid (LA) on apoptotic mechanism in a variety of cancer cells (MCF-7, DU145 and MDA-MB-453 cell lines). The authors concluded that LA induced apoptosis via miRNA-134 mediated inhibition of STAT3 and RelA/p65 acetylation. Technically the study is well done and the conclusions are supported by the data. However, a number of issues should be addressed by the authors.

qRT-PCR analyses should be detailed. The authors only have included the primers. How they analyzed data and comparisons… Also for western blots.

 (Response) Thanks for your critical comments. Added in Methods based on your comments.

LA attenuated the expression of NF-kB regulated genes in MCF-7 cells. Did the authors investigate this point in other cells?

(Response) Thanks. We published a paper entitled “Apoptotic effect of lambertianic acid throug h AMPK/FOXM1 signaling in MDA‐MB2 31 breast cancer cells” in Phytotherapy Research in 2018, where we found that LA attenuated the expression of NF-kB regulated genes such as FOXM1, XIAP, Bcl‐2 and Cyclin B1 in MDA‐MB‐231 cells.

 The effect of miR134 inhibitor on the viability of MCF-7 cells with or without LA in fig 6 seems to show that miR134 did not produce any effect on LA treated cells despite affecting pro-PARP and Ac-RelA. This point should be discussed.

 (Response) Thanks. We found statistical significance(p<0.05) between LA alone and LA+ miR134 inhibitor and added statistical significance(#) in Fig. 6C. 

This manuscript is a resubmission of an earlier submission. The following is a list of the peer review reports and author responses from that submission.